# A Scoping Review on Cue Reactivity in Methamphetamine Use Disorder

**DOI:** 10.3390/ijerph17186504

**Published:** 2020-09-07

**Authors:** Lee Seng Esmond Seow, Wei Jie Ong, Aditi Hombali, P. V. AshaRani, Mythily Subramaniam

**Affiliations:** Research Division, Institute of Mental Health, Buangkok Green Medical Park, Singapore 539747, Singapore; Wei_Jie_ONG@imh.com.sg (W.J.O.); aditihombali@gmail.com (A.H.); Asharani_PEZHUMMOOTTIL_VASUDEVAN_N@imh.com.sg (P.V.A.); mythily@imh.com.sg (M.S.)

**Keywords:** methamphetamine use disorder, cue reactivity, cue-induced craving, cue exposure

## Abstract

The experience of craving via exposure to drug-related cues often leads to relapse in drug users. This study consolidated existing empirical evidences of cue reactivity to methamphetamine to provide an overview of current literature and to inform the directions for future research. The best practice methodological framework for conducting scoping review by Arkey and O’Malley was adopted. Studies that have used a cue paradigm or reported on cue reactivity in persons with a history of methamphetamine use were included. Databases such as Medline, EMBASE, PsycINFO and CINAHL were searched using key terms, in addition to citation check and hand search. The search resulted in a total of 32 original research articles published between 2006 to 2020. Three main themes with regard to cue reactivity were identified and synthesized: (1) effects of cue exposure, (2) individual factors associated with cue reactivity, and (3) strategies that modulate craving or reactivity to cues. Exposure to methamphetamine-associated cues elicits significant craving and other autonomic reactivity. Evidence suggests that drug cue reactivity is strongly associated with indices of drug use and other individual-specific factors. Future studies should focus on high quality studies to support evidence-based interventions for reducing cue reactivity and to examine cue reactivity as an outcome measure.

## 1. Introduction

Several theoretical models of drug-use behaviors, such as the expectancy model, the dual-affect model, and the cognitive processing models have proposed that external environmental cues can serve as triggers for drug use [1,2]. Cues can produce symptoms of withdrawal in drug users, even after abstinence or detoxification [3,4]. A vast amount of empirical research has demonstrated that stimuli associated with the drug or its administration (e.g., bottle of preferred alcohol, syringe, lighter) can elicit subjective reports of craving and patterns of physiological responding in persons who have a history of drug use [5]. This phenomenon is often referred to as cue reactivity [5].

Drug cue reactivity is one of the hallmark characteristics in addiction research and numerous attempts have been made to elucidate the underlying learning mechanisms [5]. Drug cue reactivity was earlier proposed to be attributed to the formation of a direct association between the stimulus (i.e., the cue) and the response [6] but later theories began to support the view that drug cues elicit expectations of the drug, which drive drug-seeking behaviors [7]. With repeated drug experience, the drug user associates the rewarding effects of a drug with cues present at the time of consumption and this is known as classical conditioning, or Pavlovian conditioning [8]. In other words, formation of associations between cues and drugs is largely based on the premise of classical conditioning, during which initially neutral cues that are repeatedly paired with drugs (the unconditioned stimulus) acquire conditioned incentive properties and become the conditioned stimuli [7,9]. It is therefore through these Pavlovian associations that innocuous environmental stimuli become salient mediators of drug-seeking behaviors.

Cue reactivity paradigm is a valuable and versatile tool that aids the studying of cue-elicited drug craving. It typically involves the exposure of current or abstinent drug users to visual and/or auditory drug-related cues in order to monitor their reactions to these cues [10]. Exposure to drug-associated cues in laboratory settings has been shown to reliably induce drug craving and physiological reactions amongst drug users [5]. Such cue paradigms have been widely employed in addiction research across individuals who are addicted to various drugs including methamphetamine, heroin, cocaine and alcohol [5,10]. Furthermore, the application of cue reactivity paradigm has been examined in addiction research and cue reactivity studies have been proposed to offer insights into understanding the nature of drug dependence; predicting relapse and as a method of evaluating treatment efficacy [10,11,12].

Methamphetamine, a highly addictive illicit drug that is also otherwise known as ice, speed, meth and crystal, is an amphetamine-type stimulant that heightens stimulation in the central nervous system [13]. While the short-term rewarding effects include euphoria, elevated mood, reduced fatigue, increased alertness and increased libido, long-term methamphetamine use is accompanied by devastating health consequences such as having neurological abnormalities and damage, cognitive deficits, drug-induced psychosis, increased depressive and anxiety symptoms, and elevated risk of drug overdose and transmission of blood-borne diseases through sharing of needles [14,15]. Furthermore, the effects of methamphetamine were found to last longer as compared to other drugs and the drug’s high lipid solubility allows it to be transferred to the brain more readily [13]. Moreover, methamphetamine is also popularly consumed before or during sex, especially by homosexual men, to improve sexual experiences, prolong sexual performance and reduce sexual inhibition [16]. This further promotes risky sexual behaviors and the proliferation of sexually transmitted diseases amongst the drug users. More worryingly, methamphetamine has increasingly become a global public health concern It was estimated in 2017 that there were approximately 28.9 million amphetamine-type stimulant (including methamphetamine) users over the last two decades [16]. Beyond the constant evolving and expansive trends of methamphetamine manufacturing and trafficking activities across different regions of the world, methamphetamine use continues to increase in North America and Asia [16,17].

Despite amphetamine-type stimulants (mainly methamphetamine) being the second most widely used class of illicit drugs worldwide [18], no effective pharmacotherapeutic agents are available for the treatment of methamphetamine dependence, nor is there any medication approved by the regulatory authorities for such treatment till date [19]. Cue reactivity studies in methamphetamine may in turn represent a good tool of considerable utility for investigating addictive phenomena. Much research efforts in the area of cue reactivity have been made with other substances such as opiates, cocaine and alcohol, but those relating to methamphetamine appear to remain in infancy [5,10,12,20]. The current study therefore intends to consolidate and review existing empirical evidences of cue reactivity to methamphetamine so as to provide an overview on current literature and inform the directions for future research.

## 2. Materials and Methods

The current review was conducted in line with Arksey and O’Malley’s framework for scoping methodology [21] (Step 1: identify the research questions; Step 2: identify relevant studies; Step 3: study selection; Step 4: chart the data; and Step 5: collate, summarize and report the results) and the PRISMA-ScR (Preferred Reporting Items for Systematic reviews and Meta-Analyses extension for Scoping Reviews) [22] checklist was used for reporting findings.

Scoping reviews tend to have a very broadly defined research question and we therefore formulated the following: “What does cue reactivity have to offer to methamphetamine research?” to guide our review. Based on knowledge from existing literature, specific areas of applications of cue reactivity paradigms were pre-defined, but not limited to:Method of understanding the nature of methamphetamine dependencePredictor of relapseMethod of studying treatment effects

### 2.1. Search Strategy and Databases

Cue reactivity has been a frequently used method in addiction research since the 1990s with the theory and practice of cue exposure being put into perspective by Drummond (1995) [23]. An electronic search of articles published between January 1995 and November 2019 was therefore performed on EMBASE, MEDLINE (PubMed), PsycINFO and CINAHL (with full text). The main search of the databases was conducted on 15 November 2019. A prior search limit was set to include English publications and studies involving humans only. Hand searching of references from key papers and citations from the web (e.g., Cochrane Central Register of Controlled Trials, Google Scholar; last updated on 9 March 2020 to screen for new, potential studies) were also performed. Authors were also contacted for full-text articles if they were not publicly available (e.g., accepted paper but yet published) and clarification of information. Search terms (“cue” AND “methamphetamine” only) were selected to provide extensive coverage. All references were stored and managed in EndNote X7 (bibliographic software).

A two-stage review strategy—first at the title/abstract level followed by full-text level—was adopted and each potentially relevant result was examined by two authors (ES and WJ) who worked independently at each stage. Disagreements between the two authors were discussed or consulted with the third author (MS) until a consensus was reached.

### 2.2. Inclusion and Exclusion Criteria

Studies were screened and included if they had used a cue paradigm or reported on cue reactivity in persons with a history of methamphetamine use (i.e., current or previous users). Studies that were conducted in animals or among healthy participants only were excluded. In addition, studies that recruited participants with polysubstance (cocaine, cannabis, heroin etc.) use and had not reported findings specific to exposure of methamphetamine cues or the use of methamphetamine separately, were also excluded. Upon full-text screening, we decided to further exclude reviews, study protocols, extracts from books or other non-scientific publications, case reports and those that were only available as abstracts (e.g., conference or dissertation abstracts), and to include only primary studies.

### 2.3. Data Charting, Collating, Summarizing and Reporting

Data was extracted and tabulated to include the Population, Intervention, Comparison and Outcome (PICO) elements by one author (ES) and verified for completeness and accuracy by a second reviewer (WJ). Details of study population and context, cue reactivity paradigm, type of intervention and comparator (if applicable), outcome measures, and findings of interest were extracted and recorded. In order to chart the data, the studies were classified according to the broad areas of application of the cue paradigms and sorted by date of publication. Quality of evidence or formal risk-of-bias assessment for each individual study was not evaluated, as consistent with the scoping review methodology. To summarize the findings, data was synthesized and reported according to key themes. Within each broader theme, data was further sub-categorized to group common effects, factors and intervention types so as to present a more meaningful narrative account of the existing literature. All team members reviewed the themes and consensus was reached for the label of each theme.

## 3. Results

The combined search identified a total of 163 references, of which 84 were duplicates and were removed. A total of 87 citations (including an additional 8 articles found through hand search) were therefore screened on the basis of title and abstract (see Figure 1) to ensure that they addressed the appropriate population and had a focus on drug cue reactivity. 50 references did not meet our criteria at title/abstract screening and a further 5 were eventually excluded at full text-screening. Of the final 32 relevant articles accepted, 16 were from the USA, 11 were from China, 4 from Iran and 1 from Georgia. Two of these articles reported on two separate studies each, leading to a total of 34 studies being reviewed.

### 3.1. Summary of Included Studies

Visual (photographs/images and videos) (*n* = 28), in vivo paraphernalia and/or simulated drug (*n* = 10) and imagery cues (e.g., audiotaped scripts, recalling of last drug use) (*n* = 4) were the different modalities used in all the studies to elicit responses in a cue reactivity paradigm. Several studies also designed and employed the use of a methamphetamine virtual reality (methamphetamine-VR) cue model to assess self-reported cravings and physiological changes [24,25,26,27]. A total of 18 studies compared effects of methamphetamine-related cues with neutral cues; some of which involved the use of cues related to nature scenes (*n* = 4), beach (*n* = 1), images of artifacts and normal daily actions (*n* = 2), footage of tropical fish in tank/aquarium (*n* = 2) or the handling of a glass of water (*n* = 2) and pine cones, shells, and rocks (*n* = 1). Few studies also used control cues such as sexual (*n* = 2), food (*n* = 1) visual cues, as well as “happy” and “sad” stimuli based on subjective evaluations of the emotional valence of the images (*n* = 1). Only 8 studies recruited a control group with healthy participants for comparison purpose. Studies (*n* = 3) that used both control cue(s) and a control group were those that looked at difference in neural reactivity to methamphetamine vs. control vs. neutral cues using functional magnetic resonance imaging, and compared them between drug users and healthy controls. Most studies employed a single item visual analogue scale to assess cue-induced craving (*n* = 14). Established scales such as the general craving scale (GCS, *n* = 1), within session rating scale (WSRS, *n* = 4), brief substance craving scale (BSCS, *n* = 1) and methamphetamine urge questionnaire (MAUQ, *n* = 2) were also utilized to assess cue-induced drug craving. Depending on the aim of the studies, these measures were administered to participants at different time points (e.g., prior, during and after cue exposure depending on the study). Overall, this scoping review identified several themes with regard to the main applications of drug cue paradigms. As shown in Figure 2, the three main themes synthesized were: (1) effects of cue exposure, (2) individual factors associated with cue-induced cravings or other cue reactivity, and (3) strategies and measures that modulate drug craving or reactivity to cues.

### 3.2. Effects of Cue Exposure

The majority of the included studies assessed the impact of cue exposure on cue-elicited craving as the primary study outcome. A handful of studies also included more than one measure of cue reactivity such as self-reported craving with neural activation or autonomic arousal recorded at baseline, during and after cue processing (see Table 1).

#### 3.2.1. Subjective and Physiological Responses

Studies looking at cue reactivity found that methamphetamine stimuli are generally reported to increase levels of drug craving, “anxious” mood and other physiological arousal such as heart rate, blood pressure and skin conductance variability among participants who had a history of methamphetamine use. Further, two studies revealed differences in reactivity to different modalities of methamphetamine-related cues. For example, a study by Tolliver and colleagues suggested that relative to baseline, presentation of methamphetamine-associated photo, video, and paraphernalia cues elicited significant increases in subjective craving (from fewer than half of the participants at baseline to approximately 70% of participants after cue exposure) and skin conductance while heart rate was significantly decreased only after viewing methamphetamine-related photos or video, but not after exposure to paraphernalia [28]. Methamphetamine virtual reality condition was also found to induce the greatest change in subjective reports of “crave methamphetamine”, “desire methamphetamine” and “want methamphetamine” at all-time points compared to methamphetamine-video and other neutral test conditions, as well as higher increase in “anxiety” rating compared to neutral virtual reality condition [24].

#### 3.2.2. Neural Reactivity

Besides subjective and physiological responses, the scoping review also found studies examining cue reactivity with functional magnetic resonance imaging (fMRI) and electroencephalogram (EEG). Seven studies that examined fMRI evidence reported functional abnormalities in the brain of those with methamphetamine use disorder, compared to healthy participants. Blood-oxygen-level dependent measures of methamphetamine cue reactivity revealed activation of a broad set of regions, particularly the mesocorticolimbic system which includes the ventral and dorsal striatum, the cingulate cortex, prefrontal cortex (PFC) and insula [29,30,31,32,33,34,35]. Two studies looked at EEG recordings: one study utilized event-related potential (ERP) technique and found higher neural responses to drug-related visual stimuli among users compared to controls in the P300 components [36], while the other study measured gamma current density and found gamma activity in medial prefrontal cortex (mPFC)/orbitofrontal cortex (OFC) and right dorsolateral prefrontal cortex (DLPFC) to decrease after cue exposure [27].

#### 3.2.3. Cognitive Function

The cognitive effects of cue exposure in chronic methamphetamine abusers have been examined in one study. Tolliver and colleagues found the exposure of methamphetamine-related cues to impair participants’ performances (increased rates of both response errors and inhibition errors) on an auditory dual task Go–No Go cognitive test requiring divided attention and inhibition of distracting information [37].

### 3.3. Factors Associated with Drug Cue Reactivity

The scoping review identified and characterized major factors that were found to modulate craving upon presentation of methamphetamine cues in users. In this section, we have reviewed available evidence and discussed individual-specific factors that were associated with, or in some cases, predictive of cue-elicited craving (see Table 1).

#### 3.3.1. Demographics

Three studies explored the relationship between demographics and cue reactivity and reported that differences in cue-induced cravings (across individuals or cue modalities) were not associated with examined sociodemographic variables such as age, education, employment etc. [28,38,39]. These findings proposed that methamphetamine craving was largely due to cue exposure and not influenced by demographic keys. Only one study, however, found age to positively and education to negatively predict craving changes [40].

#### 3.3.2. Substance Use Profile

Findings from six out of seven studies that examined the relationship between variables relating to drug use and cue reactivity indicated significant effects of substance use-related variables on cue reactivity (except [39]). Ekhtiari et al. (2009) found age of onset of drug abuse to be negatively correlated with level of craving responsiveness [38]. Lopez et al. (2015) also revealed that the strength of cue-induced craving for methamphetamine can be moderated by users’ route of administration, such that individuals who preferred to smoke methamphetamine reported significantly stronger craving for smoking stimuli, whereas those who preferred the intranasal route reported stronger craving for intranasal stimuli [41]. The most robust predictor of cue-induced craving was found to be baseline craving for methamphetamine in the study by Tolliver and colleagues, who noted that the degree of craving at baseline was strongly associated with the frequency and amount of methamphetamine use in the 60 days prior to study entry [28]. Wang et al. (2013) also found the effects of length of methamphetamine abstinence on cue-induced craving, which increased as the length of abstinence increased until 3 months but decreased with 6 months and 1 year of abstinence. This effect was, however, not observed on cardiovascular measures including heart rate, systolic and diastolic blood pressure [42]. In terms of neural cue reactivity, Malcolm et al. (2016) similarly found the increased brain activation in the ventral striatum in response to methamphetamine cues compared to rest condition, to correlate significantly and negatively with the days since last use of methamphetamine, supporting the notion of reduced cue reactivity with longer period of abstinence [31]. Lastly, Huang et al. (2018) also found activation in the left lateral anterior cingulate cortex (ACC) region of the bilateral medial prefrontal cortex (mPFC) to methamphetamine-related image cues in users to be positively associated with previous drug use frequency [30].

#### 3.3.3. Personality Attributes

Three studies explored the relationship between personality attributes and craving in response to drug cues. Saladin et al. (2012) examined the factor alexithymia—a personality attribute characterized by a difficulty identifying and describing emotions—and predicted that it may contribute to addicted persons’ failure to report cue-elicited cravings. Contrary to their hypothesis, higher scores on the Toronto Alexithymia Scale factor 1 (a measure of difficulty in identifying feelings) were positively associated with cue-elicited craving and authors suggested that their results may need to be replicated [43]. In a study by Chen et al. (2020), both attentional and non-planning impulsivity were found to correlate negatively with methamphetamine cue-related activation among short-term (but not long-term) methamphetamine users, though these findings did not survive Bonferroni correction [35]. However, methamphetamine users who rated zero increase in craving upon cue exposure did not differ from those who rated non-zero increase in terms of impulsiveness and emotional stability [39].

#### 3.3.4. Other Cue Reactivity Responses

Several studies found an association between cue-elicited craving and other measures of cue reactivity. For neural cue reactivity, for example, three studies revealed craving response to be significantly associated with increased activation in different regions of the brain related to the mesolimbic reward pathway during methamphetamine cue processing [29,30,33]. In an attempt to examine impaired performance on a cognitive task due to the exposure of methamphetamine-related cues, Tolliver et al. (2012) found response error rates, but not inhibition error rates or reaction times, to correlate with cue-elicited craving scores in methamphetamine users [37]. For physiological reactivity, two studies found a positive relationship between changes in subjective craving response and heart rate variability measure upon exposure to cue [24,42]. In contrast, Tolliver et al. (2010) reported a lack of correlation between change in heart rate and skin conductance after cue exposure, and between either measure with cue-induced craving [28]. Tan et al. (2012) found cue-induced changes in brain electrophysiological response (gamma activity) to associate with changes in skin conductance level, but not self-reported craving while neither the change in skin conductance nor heart rate variability was correlated with craving increase [27].

### 3.4. Strategies or Interventions that Modulate Drug Cue Reactivity

We reviewed eighteen studies that assessed efficacy of measures that targeted methamphetamine cue reactivity and organized them into broad categories of “pharmacological” and “non-pharmacological” methods. Studies using “non-pharmacological” methods were further grouped into non-invasive brain stimulation or behavioral techniques (see Table 2).

#### 3.4.1. Pharmacological Measures

Five studies investigated the role of medications in the treatment of participants with methamphetamine abuse or dependence by targeting cue reactivity. These studies assessed oral naltrexone, aripiprazole and bupropion using slightly different randomized trial designs (crossover or parallel group) in clinical samples. Three studies assessed the potential effects of 50 mg naltrexone (NTX) in attenuating cue reactivity and generally found NTX to be a promising pharmacotherapy through reducing cue-induced craving or other neural and physiological cue reactivity and altering functional connectivity [33,44,45]. Furthermore, NTX was found to moderate the associations between cue-elicited craving with precuneus connectivity to the sensorimotor and frontal regions [33] and post-infusion subjective positive methamphetamine effects (i.e., good effects, feel drug, high) [45], thereby suggesting brain-activity dependent or behavioral mechanisms by which naltrexone may be efficacious in treating methamphetamine use disorder. Only one study examined the effectiveness of oral aripiprazole (15 mg) treatment, which did not reduce cue-elicited craving and cardiovascular response and the authors proposed to conduct further research with lower doses of aripiprazole before ruling it out as a treatment for methamphetamine dependence [46]. The effects of bupropion treatment were also examined in a single study and bupropion was found to block cue-induced craving [47].

#### 3.4.2. Non-Pharmacological Measures

##### Non-Invasive Brain Stimulation

Six studies assessed either repetitive transcranial magnetic stimulation (rTMS) (*n* = 4) or transcranial direct current stimulation (tDCS) (*n* = 2) of the dorsolateral prefrontal cortex (DLPFC) to evaluate changes in cue-induced craving [40,48,49,50,51,52]. Li et al. (2013) found low-frequency rTMS of 1 Hz to transiently increase cue-induced craving [48] while Su et al. (2017) found high frequency rTMS of 10 Hz to reduce cue-induced craving in methamphetamine users [40]. Despite these conflicting results with regard to the effect of rTMS on left DLPFC, it was proposed that low frequency rTMS (<1 Hz) tends to produce inhibitory effect while high frequency rTMS (>5 Hz) tends to increase cortical excitability, and therefore high frequency of 10 Hz rTMS is generally most used in the treatment of substance addiction [40]. This is also in line with the “inter-inhibition between two hemispheres” theory in neuro-rehabilitation field which proposed that excitation of unilateral cortical region leads to suppression of the contralateral side. In another words, “high frequency on the left” equals to “low frequency on the right” and therefore high frequency stimulation would result in opposing effects to low frequency treatment [53]. Yet, a later study by Liu and colleagues demonstrated conflicting results where minimal differences across the different combinations between left/right hemisphere and high/low frequency rTMS were reported and all the four protocols (10 Hz L-DLPFC, 10 Hz R-DLPFC, 1 Hz L-DLPFC, 1 Hz R-DLPFC) were found to be effective in managing cue-induced craving. The authors suggested potential explanations to their results such as the use of different rTMS modes or stimulation targets and individual differences in reaction to plasticity induction protocols [49]. A newer form of rTMS, known as the intermittent theta burst stimulation (iTBS) that can induce long-term potentiation but deliver the same number of pulses in a shorter time with excitatory effects similar to traditional 10 Hz stimulation, was also found to reduce cue-induced craving significantly over four weeks of intervention in a recent study [52]. Two studies looking at tDCS (current intensity = 2 mA) of the DLPFC also showed inconsistent results, with both revealing a state dependent effect on methamphetamine cravings (induced vs non-induced). A single session of anodal tDCS on the right DLPFC decreased immediate subjective craving at rest after 10 min but increased craving rating upon exposure to methamphetamine cues compared to the sham condition, with the more provocative cues inducing significantly more cravings [50]. On the other hand, repeated sessions of bilateral tDCS (anodal stimulation of right hemisphere and cathodal stimulation of left hemisphere) reduced cue-induced craving but did not alter instant craving [51]. It was proposed that the different numbers of tDCS sessions or montages used by the two studies may have contributed to such discrepancy in findings [51].

##### Behavioral Interventions

The efficacy of an attentional bias modification (ABM) program to reduce attention bias towards drug-related stimuli was tested in one study which indicated that the ABM training did not lead to reductions in craving for methamphetamine or in attentional bias to methamphetamine-related stimuli [34]. The authors, however, highlighted several study limitations including the need to improve measurement of attention bias, that may have led to the discouraging results of the ABM training. Three studies reported on interventions related to cue exposure therapy or counterconditioning procedure, i.e., the use of cue paradigms to extinguish conditioned responses to drug cues. DeSantis et al. (2009) found that participation in a human laboratory cue reactivity paradigm was associated with longitudinal (14 days after study) decreased odds of drug use among methamphetamine-dependent participants [54]. Price et al. (2010) also examined drug cue reactivity and response extinction in a laboratory setting where participants underwent a total of six cue exposure sequences, and data revealed a mean percentage change of −84.4% in craving score from Sequence 1 to end of the last cue sequence, with11 of the 20 participants reported no craving at the end of Sequence 6. Lastly, another study using virtual reality counterconditioning procedure (VRCP) and its computerized version found participants in the intervention groups to show a significantly larger decrease on methamphetamine craving and liking in a group of patients, as well as in heart rate variability on time domain and non-linear domain from baseline to follow-up assessments in a separate sample [26]. These results therefore indicated that extinction of drug-cue conditioned responding can occur in methamphetamine-using individuals, offering promise for the development of extinction- or counterconditioning-based treatment strategies. Findings from two studies suggested that applying behavioral self-regulation can also be helpful in reducing cue-elicited cravings. In a study by Lopez et al. (2015), participants reported significantly lower cue-induced craving when focusing on the negative consequences associated with methamphetamine use (e.g., how tired or sad they might feel the next day, how much money they spend on methamphetamine use, and any damage to their relationships resulting from use), instead of the positive aspects (e.g., how smoking methamphetamine might cause pleasant physical sensations, increase their energy, and make them feel good) [41]. In another qualitative study, respondents reported using different personalized methods in real life for controlling drug or cue-induced urges such as focusing on the importance of maintaining structure in their lives and resetting limits or obstacles that would make future use difficult [55].

## 4. Discussion

This review seeks to identify and describe the type of available literature related to cue reactivity and the use of cue paradigms in methamphetamine research. A scoping review was chosen as it helps to provide a framework to (1) map the key concepts and insights, (2) summarize and share existing research findings and (3) determine gaps in the current literature. To the authors’ knowledge, only one article published in 2010 had provided a preliminary review on studies that apply the different cues as main methods of craving induction in laboratory settings among human methamphetamine dependents [12]. Six studies [24,28,37,38,47,56] discussed in the article were also included in our scoping review. This brief report was, however, conducted ad-hoc without following a specific methodology, and may thus lack the thoroughness and rigor compared to a review that conducts its search strategy using a systematic approach.

The current scoping review identified 32 relevant articles that either involved the use of a cue reactivity paradigm in their studies or allowed us to understand more about cue reactivity in methamphetamine abuse or dependence. Our results revealed high variability in the terms of reporting and conduct of the included studies and sources of heterogeneity include differences in sample groups, methodologies, type of cue reactivity paradigms, interventions and comparators, outcome measures and even research purposes across the studies. The included articles were predominantly from two countries—USA (*n* = 16) followed by China (*n* = 11), with an under-representation of studies from other parts of the world. Importantly, three overarching themes were identified in this review and they provided us with further knowledge on (1) the effect of methamphetamine cue exposure, (2) possible factors that correlated with cue reactivity, and (3) strategies that could modulate cue reactivity.

Firstly, it is evident from our review that methamphetamine-related cues can result in significant impact associated with exposure of these cues among drug users. These include an increase in subjective craving, physiological responses, activity in specific brain regions, as well as cognitive impairment. Notably, some studies did not report any significant change in physiological reactivity upon cue exposure [24,46] while one study found a decreased heart rate in response to methamphetamine cues [28], which clearly contrasts with results from majority of cue reactivity studies in the current review, as well as for other drugs of abuse. Such variation in findings may be due to methodological heterogeneity across the studies. The increased neural reactivity in response to cues was revealed to occur mainly in the mesocorticolimbic system such as ACC, posterior cingulate cortex (PCC), DLPFC, and orbitofrontal cortex (OFC) involved in relapse mechanism, demonstrating the incentive salience of drug cues. This finding is also consistent with other neuroimaging studies conducted among individuals with other drugs such as nicotine, cocaine, heroin, marijuana and alcohol [57]. Of the studies that looked primarily at the impact of cue exposure, only seven of them had recruited healthy controls for comparison and the lack of this control group could be seen as a shortcoming of many studies. Despite so, healthy controls in all these seven studies did not show significant change in response to methamphetamine cues compared to neutral cues, thus providing support to the theory that cue reactivity may indeed be a result of Pavlovian conditioning. In other words, the lack of drug expectancies among non-drug users lead to lack of cue reactivity. Nonetheless, future studies that aimed to explore effects of cue exposure should include a healthy control group to provide further support for the theory.

The current review also identified several individual-specific and strategy-specific factors that have been shown to affect reactivity to methamphetamine-related cues (see Figure 2). The former includes methamphetamine use profile, personality attributes and the association of cue-elicited craving with neural or physiological reactivity, while the latter includes non-invasive brain stimulation (e.g., rTMS and tDCS) and behavioral techniques (e.g., cognitive regulation, cue extinction and counterconditioning procedures). Individual factors such as demographic characteristics were typically not found to associate with cue reactivity, while interventions such as aripiprazole treatment and attentional bias modification did not significantly reduce cue reactivity. Drug cue reactivity is a complex phenomenon and is not surprisingly modulated by a large number of factors (i.e., main effects) as well as their interactions. In terms of its relationship with individual-specific factors, the significance, direction and magnitude of their associations may require more robust research with larger study samples to confirm on the findings. For modulation due to interventions or strategies, the findings from studies (e.g., on aripiprazole, attentional bias modification, and cognitive regulation) with smaller sample sizes (<20 participants) in particular should be treated with caution as the likelihood of a Type II error is increased in these studies, and therefore decreases the power of the study.

### Future Directions on Treatment Options

There is currently no medication with well-established efficacy for the treatment of methamphetamine use disorder, nor is there any medication approved by regulatory authorities (e.g., U.S. Food and Drug Administration) for use in methamphetamine dependence [19]. Craving is an important symptom and server that maintains methamphetamine dependence as it is often elicited by drug-related or contextual cues, and eventually leading to relapse of the drug [58]. As a result, cue-induced craving has always been regarded as a primary target for relapse prevention.

Our review suggested that oral naltrexone (NTX), an opioid receptor antagonist, could be a potential pharmacotherapy for preventing relapse to methamphetamine. Three studies examined the efficacy of NTX in modulating or reducing cue-induced craving and all revealed positive results in favor of the drug [33,44,45]. Furthermore, NTX appeared to be well-tolerated and had very minimal side effects [44]. Despite these promising results, the use of NTX in targeting methamphetamine relapse needs to be supported by more clinical trials and evidence. Scientifically-based approaches to evaluate medications that limit brain exposure to methamphetamine, modulate methamphetamine effects at vesicular monoamine transporter-2, or target dopaminergic, serotonergic, gamma-aminobutyric acid (GABA)-ergic, and/or glutamatergic brain pathways have already been underway [19,59] and despite the increasing efforts made to review medications for the treatment of methamphetamine dependence, many of these trials failed to consider their course of action on cue-induced reactivity.

For behavioral intervention, the use of either cue extinction or counterconditioning strategies appeared to be promising in preventing relapse due to cue reactivity based on findings from our review. Studies among healthy humans have consistently shown that conditioning, the process by which contextual cues becomes associated with methamphetamine through repeated pairing, is the key to understanding addiction and problematic drug use [60,61]. Literature, however, suggests that extinction procedure may not be adequate in suppressing drug craving [62] and it was proposed that the affective component inherent to the drug-related cues tends to hinder the efficacy of cue exposure-based therapy in individuals with substance use disorder [63]. In this sense, the counterconditioning approach has been viewed to be a better alternative as it not only decreases the unconditioned expectancy, but also changes the emotional valence of conditioned stimuli through pairing new unconditioned stimuli with an opposite valence [26]. Previous studies have also supported counterconditioning procedures to exert a stronger suppressing effect on the relapse of memories or the cue-drug association than extinction [64,65]. Future methamphetamine studies could focus on these non-pharmacological strategies and compare the difference between extinction and counterconditioning procedures in reducing cue reactivity. As with research on pharmacological therapies, while literature identified other psychosocial interventions such as cognitive behavioral therapy, counselling or motivational interviewing, and contingency management to show effectiveness in the treatment of methamphetamine dependence [66], there was a paucity of research that addressed cue-induced craving and reactivity to cues as study outcomes. Lastly, there also appears to be a lack of studies looking at the combined effect of pharmacotherapy and psychological treatments in reducing reactivity to methamphetamine-related cues. 

## 5. Limitations

Although it is not the main tenet of a scoping review, it must be acknowledged that we did not formally assess the methodological quality of the studies, particularly those that evaluated interventions. Among which, we also included one qualitative study and few of these studies did not include a control group with only a pre-post study design. The majority of studies had small sample sizes except one, which had over 1000 study participants. The current review also did not include grey literature (e.g., conference abstracts), which may have led us to miss out some relevant studies or possible interventions. The restriction of searches to well-established academic databases and exclusion of grey literature may also lead to a potential publication bias as studies with null findings are less likely to be published in peer-reviewed journals.

## 6. Conclusions

Cue reactivity studies have been shown to be useful for understanding how craving would lead to continued drug-seeking behaviors and relapse among abusers in a real-life environment. Exposure to methamphetamine-associated cues can significantly induce measurable craving or other autonomic reactivity in laboratory settings. Our scoping review provides insights into the type of cue reactivity, as well as in identifying and characterizing specific factors that modulate this reactivity in methamphetamine research. The use of cue reactivity paradigms also has important implications for the development of new pharmacological and psychosocial interventions for methamphetamine relapse prevention. Further studies on cue-induced craving are necessary to explore the effects that this notion could bring to treatment approaches.

## Figures and Tables

**Figure 1 ijerph-17-06504-f001:**
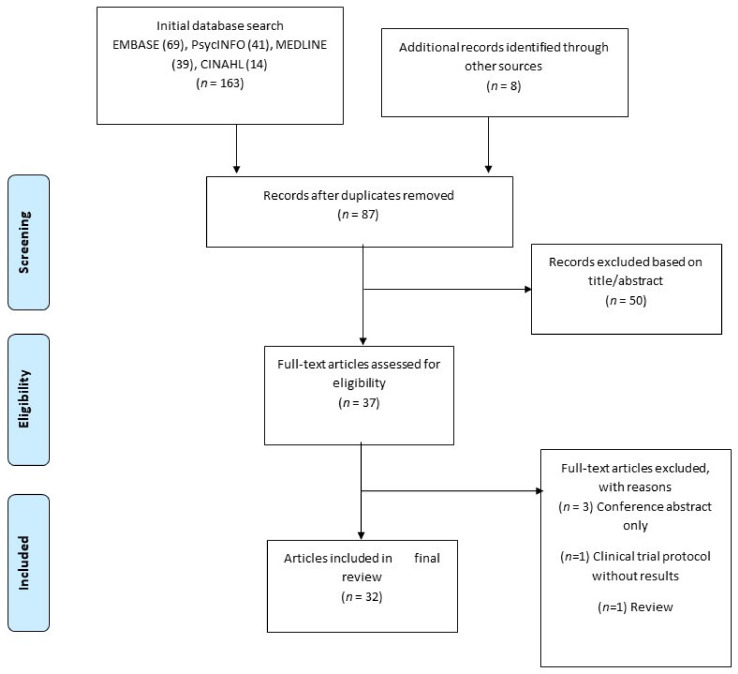
Summary of search strategy.

**Figure 2 ijerph-17-06504-f002:**
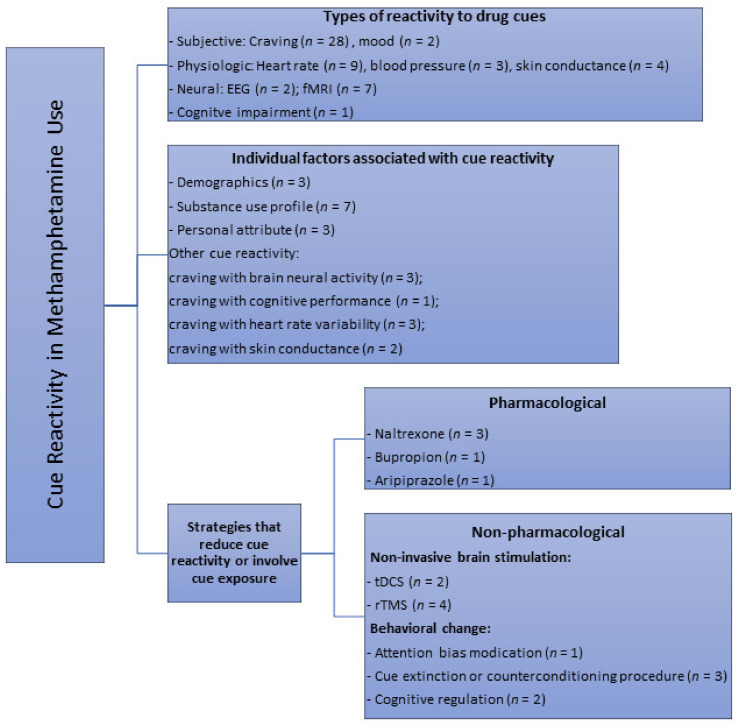
Key themes of cue reactivity or areas of cue paradigm applications identified from primary studies in methamphetamine research through the scoping review. *n* = number of available studies.

**Table 1 ijerph-17-06504-t001:** Details of included studies that looked at the effects of drug cue exposure and factors associated with change in cravings upon cue exposure.

Authors (Year)	Study Sample & Context	Drug Cue(s)	Methods	Outcome(s) of Interest	Main Findings
Culbertson et al. (2010)	17 non-treatment seeking METH users; USA	METH virtual reality (METH-VR) of virtual avatars and drug-use animations of smoking, injecting, snorting) created using online gaming platform; METH-video that included professional actors/actresses administering METH followed by in vivo mock METH paraphernalia	Participants were asked to complete four test sessions: (1) METH-VR, (2) neutral-VR, (3) METH-video and (4) neutral-video in a counter-balanced fashion.	- Subjective ratings of urges to use METH, mood, and physical state on a VAS (1-100, “none” to “very much”) prior to (time = 0), during (time = 5), after (time = 10) and following (time = 15) each cue condition - Heart rate variability (HRV) recorded over 10 min interval	- METH-VR cue condition elicited greater increases in subjective cravings (VAS) compared to all neutral cue conditions. - Participants also reported higher increase in “anxiety” (VAS) to METH-VR compared to neutral VR. No effect of cue condition on HRV measures was found. - “high craving” and ‘low craving” participants tend to display more high and low frequency cardiovascular activity (HRV), respectively during the cue conditions.
Ekhtiari et al. (2010)	50 outpatients who met DSM-IV-TR criteria for METH dependence in the past 6 months; Iran	Cues were classified into 4 main themes (drug, instruments ^1^, accompanying cues ^2^ and act of abuse) and photos were taken for each cue ^1^ refers to drug paraphernalia ^2^ refers for example to candies, beverages, money etc.	50 photos with high levels of evocative potency (CICT 50) and 10 photos with the most evocative potency (CICT 10) out of 72 cues (60 active evocative photos + 10 neutral photos) were rated by participants.	- Self-reported craving intensity on VAS (0-100) when presented with cues	- Differences in cue-induced craving (VAS) in CICT 50 and CICT 10 were not associated with age, education, income, marital status, employment and sexual activity in the past 30 days prior to study entry. - Family living condition was marginally correlated with higher scores in CICT 50. Age of onset for (opioids, cocaine and methamphetamine) was negatively correlated with CICT 50 and CICT 10 and age of first opiate use was negatively correlated with CICT 50.
Tolliver et al. (2010)	43 treatment and non-treatment seeking participants who met DSM-IV criteria for METH dependence in the past 6 months; USA	(1) 30–35 still photographs of individuals procuring and using METH, (2) a 7–8 min video depicting METH use in a variety of settings, and (3) in vivo paraphernalia and simulated METH placed in front of participants for 5 minutes	Participants were exposed to the three cue modalities in a counter-balanced fashion. Clinical and demographic correlates of METH craving were also explored.	- Subjective craving on WSRS using 100 mm VAS anchored with adjectival modifiers (not at all. mildly, moderately, extremely) - Physiological responses such as heart rate and skin conductance - All measures were obtained during and immediately after exposure to each cue modality - Baseline measures were collected 20-min and 5-min prior to the cue exposure	- Relative to baseline, subjective craving (WSRS-VAS) was increased by all three cue modalities to a similar extent. - Physiological cue reactivity correlated poorly with cue induced craving. - Differences in cue-induced craving (WSRS- VAS) were not associated with age, sex, education, employment, treatment status, or number of days using METH in the 60 days prior to study entry.
Saladin et al. (2012)	40 treatment and non-treatment seeking participants who met DSM-IV criteria for METH dependence; USA	(1) 30–35 still photographs of individuals procuring and using METH, (2) a 7–8 min video depicting METH use in a variety of settings, and (3) in vivo paraphernalia and simulated METH placed in front of participants for 5 minutes	The relationship between alexithymia and baseline and cue-elicited craving was examined.	- METH craving on WSRS using 100 mm VAS anchored with adjectival modifiers (not at all, mildly, moderately, extremely) after each cue presentation	- Toronto Alexithymia Scale factor 1 (a measure of difficulty identifying feelings) scores measured at baseline were found to positively associate with cue-elicited craving (WSRS-VAS).
Tolliver et al. (2012)	30 participants who met DSM-IV criteria for METH dependence in the past 6 months and 30 controls; USA	Public domain video footage with sequential 15–30 s segments of individuals or actors manufacturing, procuring, or using METH through various routes of administration	Participants were instructed to perform an auditory dual task cognitive test while viewing METH-related and neutral video cues in a counter-balanced fashion.	- Subjective craving on WSRS recorded before and after each video cue presentation - Reaction time, response errors, and inhibition errors on the auditory Go–No Go task	- Both response errors and inhibition errors increased significantly in METH participants while control participants exhibited only slightly increased rates of response errors upon exposure to cues. - Only response error rates, during exposure, were significantly associated with craving scores (WSRS) in METH participants.
Yin et al. (2012)	26 METH users who had not received recent treatment and had not taken the drug at least 24 h before the experiment and 26 gender-matched controls; China	Not clearly described	Participants viewed METH cues vs neutral cues vs happy and sad (control cues) stimuli based on subjective evaluations of the emotional valence of the pictures via a block design in a balanced order.	- Activation in specific affect-related regions of the brain (ACC and frontal gyrus) were recorded with fMRI while exposure to picture cues	- Robust activation of the ACC gyrus was evident in patients watching METH cues, but not in those watching sad or happy pictures or in healthy participants under any condition. - In contrast, patients showed less activation than healthy participants during the METH-cue pictures in areas of the frontal lobes.
Wang et al. (2013)	139 inpatients who had a history of DSM-IV METH dependence with varying abstinence period: 6 d (*n* = 24), 14 d (*n* = 26), 1 m (*n* = 19), 3 m (*n* = 20), 6 m (*n* = 20) and 1 y (*n* = 29); China	(1) 32 photographs of individuals procuring and using METH, with each presented for 7 s in a slide show, (2) a 5 min video in which METH abusers made drug paraphernalia and used METH, and (3) paraphernalia placed directly on the table in front of the participants	Participants underwent a cue session where either neutral cues or METH cues were presented first in a randomized manner.	- Subjective craving on VAS (1–10, “not at all” to “extremely high”)- Physiological responses such as heart rate and blood pressure- All measures were recorded before and after each cue presentation	- Cue-induced craving (VAS) increased until 3 months of abstinence and decreased at 6 months and 1 year of abstinence. - The effect of length of abstinence on cue-induced physiological measures did not differ significantly.
Lopez et al. (2015); Study 1	21 participants who met criteria for METH abuse or dependence as assessed by SCID-IV; USA	84 images and 42 short videos from online sources, documentaries and feature films depicting METH use	Participants were grouped by their preferred route of administration (intranasal vs. smoking) and were shown visual stimuli- food (control cues) vs. people smoking METH vs. people snorting METH vs. ’substance only’ with no specific route of administration.	- Level of craving on a single-item scale (1–5 rating scale, “not at all” to “very much”) assessed after presentation of picture cues	- Participants who preferred to smoke METH reported significantly stronger craving for smoking stimuli, whereas those who preferred the intranasal route reported stronger craving for intranasal stimuli. - Meth users reported significantly higher craving for all METH stimuli to food stimuli. METH smokers and intranasal users did not differ in reported craving for food.
Malcolm et al. (2016)	9 non-treatment seeking males who met DSM-IV-TR criteria for current METH dependence and 9 gender-, race- and alcohol use-matched controls; USA	Sequence of slides consisting of meth use (IV, nasal, smoking) pictures	Participants viewed visual cues of METH, neutral objects (matched for color and hue) and rest (crosshair on a neutral background) conditions in a randomized presentation.	- Activation of brain circuitry recorded with fMRI while presented with cues - Current urge to use METH (0–4 scale) after each blocks of stimulus categories	- METH participants rated their craving for METH cues significantly higher and had increased brain activation in the ventral striatum and medial frontal cortex compared to controls in response to METH cues (vs. neutral cues). - The ventral striatum activation was found to correlate significantly and negatively with the days since the last use of METH among the dependents.
Shahmohammadi et al. (2016)	10 pure METH users for at least 6 months and 10 age- and gender-matched controls; Iran	Color photographs depicted on a black background containing drug associated cues and drugs (including with face and hand)	Participants viewed a series of images with neutral and METH-related content presented in fixed pre-randomized order	- Event-related potentials recordings while presented with cues	- Drug abusers exhibited significant positive activities in response to METH-related cues, which are most pronounced as P300 peaks in time range from about 300 to 600 ms and were maximal in channels FP1, FPz, FP2 and F8.
Huang et al. (2018)	28 participants who met criteria for METH dependence as assessed by SCID-IV after long term (>16 months) drug rehabilitation and 27 age-matched controls; China	30 images that fall into METH sample, drug paraphernalia and simulation scenarios of drug use shot by researchers	Participants viewed visual cues of METH, sexual (control cues), and neutral cues with order of images, blocks within epoch, and the epochs all randomly presented.	- Patterns of cortical activation recorded with fMRI while presented with cues - Subjective drug craving on VAS (0–10, “weakest craving” to “strongest craving”) assessed prior to and immediately following each MRI scan	- Elevated activity in the bilateral mPFC and right lateral posterior cingulate cortex in response to METH cues was observed among those with METH use disorder compared to controls. - Activation of the anterior cingulate region of mPFC was positively correlated with change in craving scores (VAS) and previous drug use frequency. - Compared to METH cues, those with METH use disorder had increased activation in the occipital lobe when exposed to sexual cues.
Wang, Shen and Wu. (2018)	61 male patients who met DSM-IV criteria for METH dependence and had completed more than 1 month of forced detoxification and 45 age-matched male controls; China	A METH-related virtual social environment created VR video depicting a real-life story of men/women who are using METH together and who invite the observers to take METH	Participants first went through 8-min resting state, followed by 8-min viewing of the METH-cue video.	- Cue-induced cravings on VAS (0-10, “no craving at all” to “extremely strong craving”) assessed immediately after the cue - Heart rate variability (HRV) recorded with ECG	- Cue-induced condition elicited a larger HRV in patients with METH dependence, whereas a reverse pattern of HRV change was observed in the controls. - Among the patient group, subjective craving scores were associated with HRV changes.
Grodin, Courtney and Ray. (2019)	15 non-treatment seeking participants with current METH use disorder; USA	32 METH cue pictures consisting of drug, drug pipes, and drug use	Participants completed METH Infusion in the laboratory before completing two runs of cue task, which included 4 blocks of METH cues and 4 blocks of neutral cues presented pseudo randomly. Relationship between METH-induced craving and neural responses to METH cues was explored.	- Neural responses or patterns of brain activation recorded by fMRI while presented with cues	- METH cues activated a widespread set of regions, including mesocorticolimbic regions, such as the ventral and dorsal striatum, and ventromedial prefrontal cortex, compared to neutral cues. - Higher activation to METH cues was also observed in the precuneus, insula, anterior and posterior cingulate, and occipital lobe. - Peak ratings of METH-induced craving was associated positively with neural responses in the precuneus, putamen, and ventromedial prefrontal cortex.
Liang et al. (2019)	52 males who met DSM-5 criteria for METH use disorder; China	METH use-related pictures consisting of intake utensils, tools and the scenarios of intake	Participants were exposed to six METH-related images presented in a block-wise method for 24 s (4 s each). The reliability of cue-induced craving as an indicator for addiction severity was examined.	- Cue-induced craving on VAS (0 -100, “not at all” to “extremely intense”) assessed after viewing picture cues	- 24 of the 52 METH users rated non-zero increase in subjective craving (VAS) upon exposure to cues. - Those who rated non-zero were not distinct from users who rated zero in terms of age, impulsiveness, emotion stability and clinical characteristics of addiction severity including MA use duration, maximum amount and weekly amount.
Tan et al. (2019)	60 males who met DSM-5 criteria for METH use disorder; China	A METH-related virtual social environment designed based on results of focus group, depicting a video in which four persons played with paraphernalia, smoked and talked about the quality of METH crystals	Participants were exposed to neutral environment (5 min) and drug cue environment (5 min) presented in VR headset sequentially.	- Self-reported craving on VAS (0-10, “no craving” to “most craving ever experienced for METH”) assessed after presentation of cue) - Physiological responses such as skin conductance and heart rate variability (HRV) recorded while presented with cues - Brain electrophysiological response (gamma activity) recorded with EEG while presented with cues	- Self-reported craving (VAS) and skin conductance level increased in response to VR drug cues compared with neutral cues. - HRV was only marginally increased but not significant. - Gamma activity in mPFC/OFC and right DLPFC were decreased after cue exposure and predicted the ski conductance level changes. - Self-reported craving (VAS) was not associated with electrophysiological or physiological responses.
Chen et al. (2019)	99 males who met criteria for METH dependence as assessed by SCID-IV; of which 49 men had short term (1–3 months) while 50 had long term (16–40 months) abstinence and 47 controls; China	30 images of METH itself, people who were smoking METH, or the instruments they used to smoke METH	Participants viewed visual cues of METH, sexual (control cues), and neutral cues with order of images, blocks within epoch, and the epochs all pseudo-randomly presented. Relationships between regional activations and baseline methamphetamine use and impulsivity were also explored.	- Regional brain activations recorded with fMRI while presented with cues	- Greater METH cue–related activation in the ventral mPFC was observed in METH-using participants relative to healthy controls. - METH users also displayed greater sexual cue-related anterior insula activation compared to METH and neutral cues, with no difference reported between short- and long- term abstinence groups in anterior insula responses. - In short-term METH abstinence participants, both attentional and nonplanning impulsivity scores negatively correlated with METH cue–related superior frontal cortex activation.

ACC: anterior cingulate cortex; CICT: cue induced craving assessment task; DLPFC: dorsolateral prefrontal cortex; EEG: electroencephalogram; fMRI: functional magnetic resonance imaging; IV: intravenous; METH: methamphetamine; mPFC: medial prefrontal cortex; OFC: orbitofrontal cortex; SCID-IV: structured clinical interview for DSM-IV; VAS: visual analog scale; WSRS: within-session rating scale.

**Table 2 ijerph-17-06504-t002:** Details of included studies that looked at methods that modulate cue-elicited cravings and reactivity or involve cue exposure.

Authors (Year)	Study Sample & Context	Study Design	Drug Cue(s)	Methods	Outcome(s) of Interest	Main Findings
**Pharmacological methods**						
Newton et al. (2006)	20 non- treatment seeking participants who met DSM-IV-TR criteria for METH abuse or dependence; USA	Double-blind, placebo controlled, parallel group design	Videotaped cues showing actors using METH	Following baseline METH dosing, participants received a second identical series of METH doses 6 days after initiation of twice- daily oral 150 mg bupropion (*n* = 10) or placebo (*n* = 10) before cue exposure session.	- Cue induced cravings on GCS and WSRS - Both scales administrated twice before randomization and twice after randomization	- Treatment with bupropion, compared to placebo, was associated with significantly reduced cue-induced craving on the GCS Total Score and in the Behavioral Intention subscale of the GCS. - Similar results were obtained for WSRS ‘likely to use’ but not WSRS ‘feel like using’.
Newton et al. (2008)	16 non-treatment seeking participants who met DSM-IV-TR criteria for METH dependence; USA	Double-blind, placebo controlled, parallel group design	5-min of METH paraphernalia (pipe stems, a lighter, and a small plastic bag containing white powder) viewing and handling followed by 10-min of video (actors using METH) viewing	Following baseline METH dosing, participants received repeated METH dosing after 2-week treatment with oral 15 mg aripiprazole (*n* = 8) or placebo (*n* = 8) before cue-exposure session.	- Cue-induced craving on BSCS before and after cue presentation - Subjective effects: “desire for METH”, “depressed”, “anxious”, “stimulated”, “likely to use METH” and “METH-like effect” measured on VAS before and after cue presentation - Physiological responses such as blood pressure and heart rate assessed in 5-min intervals before, during and after cue presentation	- No significant effects of aripiprazoletreatment on cue-induced METH craving (BSCS) was observed, although exposure to METH cues induced moderate increases in craving. - VAS measures on “anxious”, “nervous” and “irritable” were higher in group receiving aripiprazole both pre- and post- cue exposure. - There was no effect of aripiprazole treatment, or cue exposure on blood pressure and heart rate.
Ray et al. (2015)	30 non-treatment seeking participants who met DSM-IV criteria for METH abuse or dependence; USA	Double-blind, randomized, crossover, placebo-controlled	Audiotaped script that induced sensory and emotional memories related to METH use and the handling of METH paraphernalia (e.g., glass pipe) at various times of exposure	Participants completed two separate 5-day inpatient stays. During each admission, participants completed testing sessions comprised of METH cue reactivity and intravenous 30 mg METH administration after receiving oral 50 mg Naltrexone or placebo for 4 days.	- Cue-induced craving on MAUQ assessed after each standardized exposure - Physiological responses such as heart rate and blood pressure assessed before and after cue administration	- Naltrexone was found to reduce cue-induced craving (MAUQ), as compared with placebo. - Significant increase in heart rate and diastolic blood pressure during the METH cue compared to control cue was reported in the placebo condition, but these effects were not significant in the Naltrexone condition.
Courtney, Ghahremani and Ray. (2016)	23 non-treatment seeking participants who met DSM-5 criteria for METH use disorder; USA	Randomized, placebo controlled, within-subject design	4 blocks of METH cue pictures, with each block consisting of four pictures, presented for 5 s each	Participants underwent a cue reactivity task during two fMRI sessions following 3 days of 50 mg naltrexone administration and matched time for placebo.	- Blood-oxygen-level dependent activation and functional connectivity recorded with fMRI while presented with cues - Subjective craving on a urge scale (1–4, “no urge” to “high urge”) assessed following each block of picture cues	- Administration of naltrexone reduced cue reactivity in sensorimotor areas and occipital regions and was associated with altered functional connectivity of dorsal striatum, ventral tegmental area, and precuneus with frontal, visual, sensory, and motor-related regions. - Naltrexone weakened the associations between subjective craving and functional connectivity with sensorimotor regions but strengthened its associations with dorsal striatum and frontal regions connectivity, thus engaging greater frontal regulation over salience attribution.
Roche et al. (2016)	30 non-treatment seeking participants who met criteria for METH abuse of dependence as assessed by SCID-IV; USA	Randomized, counter-balanced, and double-blind	Audiotaped script that induced sensory and emotional memories related to METH use and the handling of METH paraphernalia (e.g., glass pipe) at various times of exposure	Participants completed two 4-day medication regimens of oral 50 mg naltrexone or placebo. On day 4 of each medication regimen, they completed a cue reactivity paradigm followed by intravenous METH administration.	- Cue-induced craving on MAUQ assessed after each standardized exposure	- Cue-induced craving for METH (MAUQ) was positively associated with post-infusion subjective METH effects, including positive, negative and craving-related responses. - Naltrexone (vs. placebo) significantly reduced the association between cue-induced craving (MAUQ) and positive subjective response to METH.
**Non-Pharmacological methods**						
Bruehl et al. (2006)	82 active METH users; Georgia	Qualitative	Not applicable	In-depth interviewing	- Narrative responses that corresponded with three types of craving (cue-, drug- and withdrawal-induced)	- Traditional cues, drugs and withdrawal states may lead to craving but do not necessarily provoke it. - Users described being able to overcome craving through personalized methods of control.
DeSantis et al. (2009)	40 participants who met DSM-IV criteria for METH abuse and dependence in the past six months but maintained abstinence on test day; USA	Longitudinal	(1) 30–35 still photographs of individuals procuring and using METH, (2) a 7–8 min video depicting METH use in a variety of settings, and (3) in vivo paraphernalia and simulated METH placed in front of participants for 5 min	Participants underwent a human laboratory cue exposure procedures.	- Subjective reports of craving (unclear on the scale used) - Physiological responses such as skin conductance and heart rate - All measures were assessed before, during and immediately after exposure to each cue modality. - Dollar value and frequency of METH use for 90 days prior and 14 days following the study as assessed by TLFB (not cue reactivity).	- Participation in cue reactivity paradigm decreased the odds (OR = 0.39) of remaining in or transitioning to the high use state, though not significant. - None of the 25 participants who for whom follow-up data were available used METH in the two weeks after participation in the study. *Results on cue reactivity were not reported.
Price et al. (2010)	24 participants who met DSM-IV criteria for METH dependence in the past six months; USA	Pre-post design	Pictures and video of individuals procuring and using METH and “in vivo” paraphernalia and simulated METH	Participant underwent 20-min sequences of multi-modal METH cue exposure over each of two one-hour sessions (total of six cue sequences), with multi-modal METH cues counter-balanced for presentation order.	- Cue-induced craving on modified WSRS-VAS (0–10) - Physiological responses such as heart rate and skin conductance - All measures were collected 20-min and 5-min prior to initial cue exposure for each session and subsequently during each cue sequence	- METH cue-elicited craving (WSRS-VAS) was extinguished during two sessions of repeated within-session exposures to multi-modal cues, with no evidence of spontaneous recovery between sessions. - No significant changes were identified for heart rate and skin conductance patterns. - A greater decrease in conditioned craving was observed in the group with longer (4–7 days) inter-session intervals, compared to those with ≤3 days.
Li et al. (2013)	10 non-treatment seeking participants who met DSM-IV-TR criteria for METH dependence and 8 gender-, race-, and other biographical characteristics-matched controls; USA	Single-blind, crossover, sham-controlled	40 METH-related pictures (drug, paraphernalia, or persons using the drug)	Participants were randomized to receive 15 min of sham and real (1 Hz) DLPFC rTMS in two experimental sessions separated by 1 h.	- Cue-induced craving on VAS (100 mm lines with anchoring statements at both ends, “no craving at all” to “the most craving I have ever had”) at baseline and during stimulation	- Real rTMS over the left DLPFC increased self-reported craving (VAS) as compared to sham stimulation in METH users, but no effect on craving in control group was observed.
Shahbabaie et al. (2014)	32 male who met DSM-IV criteria for METH dependence for a history of at least 12 months and were abstinent from any drug use for a least a week prior to experiment (mean abstinence duration= 73.33 days); Iran	Double-blind, crossover, sham-controlled	Computerized cue-induced craving assessment task comprising of two series of 20 drug related images each	20 min ‘anodal’ tDCS (2 mA) or ‘sham’ tDCS was applied over right DLPFC in a random sequence while participants performed a craving task starting after 10 min of stimulation.	- Self-reported craving on VAS (0–100 scale) before tDCS, after 10 min of tDCS, and after tDCS termination	- Active prefrontal tDCS increased craving (VAS) upon meth-related cue exposure. - The more provocative picture cues (drugs> drug use process> instruments> associated cues) induced significantly more craving (VAS) in the active condition in comparison to the sham condition.
Lopez et al. (2015); Study 2	13 METH smokers who met criteria for METH abuse of dependence as assessed by SCID-IV; USA	Within subject design	54 METH stimuli that were rated at least 3.5 on a 1 to 5 scale on their effectiveness in eliciting craving	METH smokers implemented cognitive regulation (either focusing on positive or negative consequences or no regulation) while viewing photographs depicting METH smoking.	- Level of craving on a single-item scale (0–10, “not at all” to “very much”) at the end of each trial	- Participants reported significantly lower craving when focusing on the negative consequences associated with METH use.
Liu et al. (2017)	50 male pure METH abusers; China	Randomized, sham-controlled	Handling with tools of drug use and faked METH for 5 min	Participants were randomized to receive five modes of rTMS stimulation: 10 Hz left P3 (sham), 10 Hz L-DLPFC, 10 Hz R-DLPFC, 1 Hz L-DLPFC, 1 Hz R-DLPFC.	-Cue-induced craving on a 0–100 scale prior to rTMS stimulation, 30 min after rTMS on day 1, and 30 min after on day 5	- Both high and low frequency rTMS at either left or right DLPFC decreased cue-induced craving score immediately and after 5 days of continuous treatment; while no such effect was observed for active rTMS stimulation at P3 point.
Su et al. (2017)	30 males who met DSM-5 criteria for moderate or severe METH use disorders; China	Randomized, double blind and controlled clinical trial	80 MA-related (drug-use materials, person and situation) pictures and recalling of last use of METH	Participants were randomized to receive 5 sessions of 8 min sham (*n* = 15) or 10 Hz rTMS (*n* = 15) to the left DLPFC.	- Cue-induced craving on VAS (0–100 mm, “no craving” to “most craving ever experienced for MA”) before and after real rTMS or sham stimulation as well as pre experiment baseline	- Real rTMS over the left DLPFC reduced craving (VAS) significantly after 5 sessions of rTMS as compared to sham stimulation. - Changes in craving ratings (VAS) were also significantly predicted positively by age and negatively by education.
Rohani Anaraki et al. (2019)	30 male who met DSM-5 criteria for methamphetamine use disorder and were abstinent from any drug use for at least one week before treatment (mean abstinence duration = 57.46 days); Iran	Randomized, double-blind, sham-controlled	Verbal induction where participants were asked to describe 3 previous situations that had led to craving and drug use	Participants underwent 5 sessions of 20 min bilateral real (*n* = 15) or sham (*n* = 15) 2 mA tDCS (anode right/cathode left) of DLPFC.	- Cue-induced craving on VAS (0–100 mm, “I absolutely don’t have a craving” to “It is the strongest craving I have ever had”) at pretest and posttest	- Cue-induced craving (VAS) was reduced significantly in tDCS related to sham condition.
Dean et al. (2019)	17 participants who met DSM-IV criteria for current METH dependence who were receiving residential treatment; USA	Randomized, single blind controlled trial	METH-related images consisting of glass pipes, METH in crystallized or powered form, people smoking METH (without faces shown), or any combination of these	Participants were randomly assigned to either receive 12 sessions of computerized attentional bias modification (ABM) training (train attention away from METH stimuli 100% of the time) (*n* = 8) or an attentional control condition (away from METH stimuli 50% of the time) (*n* = 9).	- Cue-induced craving on a Likert-type scale (0–4, “not at all” to “very much”) following cue presentation- Brain activation recorded by fMRI when presented with cues	- Cue-induced cravings and activation in the ventromedial prefrontal cortex was reduced over time, but ABM training did not influence these effects.
Wang, Liu and Shen. (2019)	Study 1: 61 male patients who met DSM-IV criteria for METH dependence; China Study 2: 1008 abstinent participants with a history of METH dependence recruited from 4 detoxification centres; China	Randomized controlled trial	VR METH-cue model comprising of 8-min VR video, which simulates a real METH-related social context including various METH-related cues For counterconditioning procedure, participants also viewed the characters in the videos experience a distinct adverse consequence caused by METH use.	Study 1: Participants were randomly assigned to either the intervention group (*n* = 31) who received VRCP or the waiting list group (*n* = 29) who did not receive VRCP. Study 2: Participants were assigned into intervention group (*n* = 643) and waiting-list group (*n* = 305). The former group received the computerized VRCP, while the latter did not.	Study 1: - Cue-induced ECG assessed concurrently under the exposure to VR cues - Three subjective scores on the extent of how participants crave Meth right now, find Meth pleasant/unpleasant and are likely to use Meth if they have access on VAS (0–10) after VR cue Study 2: - Cue-induced ECG assessed concurrently under the exposure to VR cues	- Study 1: Those who received VRCP showed a significantly larger decrease on the score of METH-craving and METH-liking from baseline to follow-up assessments, compared to those who did not received VRCP. - Study 1 and 2: Participant in the intervention group (those who received VRCP_ showed a significantly larger decrease in HRV indexes on time and non-linear domains from baseline to follow-up assessments upon exposure to VR cues, compared to those in waiting-list group.
Su et al. (2020)	126 treatment seeking participants who met DSM-5 criteria for severe METH use disorder; China	Randomized, double-blind, sham-controlled	5-min view of METH-related pictures	Participants were randomized to receive either intermittent theta burst stimulation (iTBS; *n* = 70) or sham (*n* = 56) over the DLPFC for four weeks (20 daily sessions, 900 pulses per day).	- Cue-induced craving on VAS (0–100, “no craving” to “highest craving for METH”) at baseline and each of the 5 study sessions (Week 1–4)	- After four weeks of intervention, cue-induced craving rating (VAS) showed a significant time × group interaction effect and a significant difference for time. - iTBS reduced cue-induced craving (VAS) whereas sham did not.

BSCS: brief substance craving scale; DEQ: drug effects questionnaire; DLPFC: dorsolateral prefrontal cortex; ECG: electrocardiogram; GCS: general craving scale; MAUQ: methamphetamine urge questionnaire; METH: methamphetamine; tDCS: transcranial direct current stimulation; TLFB: time-line follow-back; rTMS: repetitive transcranial magnetic stimulation; VAS: visual analog scale; VRCP: virtual reality counter-conditioning procedure; WSRS: within-session rating scale.

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
