# Peer review of "A Scoping Review on Cue Reactivity in Methamphetamine Use Disorder"

_ijerph, 2020, doi:10.3390/ijerph17186504_

Round 1

Reviewer 1 Report

Overall feedback:

While I enjoyed the subject matter of the review, there is a need for major restructuring of the writing for this to be citable/useful for other researchers. This is mainly due to substantial lack of detail in almost every section of the review. There is no way to understand how reliable the studies presented are, and I am no better off understanding what cue reactivity provides in the literature at the end of reading the review (does cue reactivity even predict drug use? What is the significance?). The authors are taking the claims of the studies at face value, and it is unclear whether the cue reactivity is studied in casual meth users or people with substance use disorder. The formatting inconsistencies in the tables also compound this problem of lack of detail. I am aware that scoping review need not provide specific answers. However, it is unclear what the questions are and what the review is trying to achieve (even though the review states it, the writing itself does not match such aims). All the inconsistent findings in seemingly ‘similar’ studies are all brushed off as due to ‘methodological heterogeneity’ – but how were they heterogeneous? Which study had what sort of control groups and stimuli? Specific suggestions to improve each section are below.

Abstract first sentence and throughout the manuscript including in table:

  1. Do not refer to drug users or people with substance use disorder as ‘addicts’ - Even if the original study used the word ‘addicts’, it has been found to carry stigma and professionals are encourage to not use that word. You can write meth users or people with meth use disorder.

Introduction:

  1. Introduction is missing a lot of references, which raises doubts on whether craving/cue reactivity contributes towards drug use. Every sentence should be referenced in the first paragraph. Also, are there studies actually linking craving to drug use, measured by urine tests? These empirical evidence is necessary to first establish cue reactivity is worthwhile to understand.
  2. First paragraph – describe DSM-IV definition of craving, or state whether craving is also defined in DSM-IV. This is important because many studies described in the review used this DSM-IV rather than DSM-5 (also, consistently use DSM-5 not DSM-V throughout the manuscript).
  3. Devote one or two paragraphs (with full references) on where cue reactivity in meth users may come from, which is critical in any manuscript attempting to use cue reactivity to understand substance use. Conditioning is what people propose (Volkow et al., 2006), but for cue reactivity to be due to emotional learning such as Pavlovian conditioning, it has to be more than a. control stimulus (i.e., specific to the emotion), and such discrimination has to be more than b. control participants that never had meth using experience (i.e., specific to the experience). Many cue reactivity studies actually do not meet such conditions to claim cue reactivity, and that’s one of the ‘methodological heterogeneities’ the authors need to highlight in the introduction so that the readers can determine which studies to follow for their own studies.
  4. Please reference Line 57: “Despite being the 2nd most common illicit drug abused worldwide”; Line 59: “Methamphetamine craving can begin within a few hours to a few days after abstinence”.

Materials and methods:

  1. Line 81: state the rational behind the Jan 2005 cut off.
  2. Line 96: “persons with a history of methamphetamine use disorder (i.e., current or previous users)” – can you specify whether there was a formal diagnosis of meth use disorder? It seems a lot of the studies included didn’t. Maybe best to rephrase as “a history of meth use”. Meth use is very different from meth use disorder.

Results:

  1. Figure 2- remove the asterisks in the figure and the caption. It seems inconsistent with describing the rest of the studies in this particular figure, not all the studies have been determined that way in this figure.
  2. Tables are not very informative as they are. Table 1 needs to include more columns and be split into two. First one can focus on the study details of all studies included in the review (can still specify design of non-clinical trials). You need to add 1) a column for meth users detailing sample size and the diagnostic criteria used (ICD-10 or DSM-IV or DSM-5 or none – if DSM-IV was it abuse or dependence or both?); 2) another column detailing control group (e.g., age- or sex-matched, sample size (can be none)); 3) control cue column detailing what they were (e.g., you can’t write ‘neutral cue’ when pornography was used, you have to specify ‘pornography’ as control cue); 4) when the outcomes were measured; and 5) exactly what scale was used in the outcomes. Table 2 can then describe the findings of each study. If outcomes had subjective (i.e., self-report) and/or objective (i.e., heart-rate or skin conductance) measures, specify the findings for every different type of measure.
  3. Table 1: In method it is stated that they used Population, Intervention, Comparison and Outcome (PICO), however this is not how the table is presented.
  4. Table 1: In Yin 2012, pictures of happy/sad what? (faces?)
  5. Lopez et al 2015 – what is the DSM-IV diagnosis? Dependence or abuse?
  6. Overall, formatting of the tables is very inconsistent, really difficult to glean important information with such inconsistencies. An example in Table 2 is Su et al 2017. The population is listed as “30 males who met DSM criteria for meth use” whereas right below in Anaraki et al 2019 it says: “30 abstinent meth users (DSM-5)”. The authors should revise to format it more consistent and clearly. Table 2: At times they list the dose, then medication, with sample size in brackets, other times they put the medication dose in bracket and no sample size.
  7. Roche et al 2016: DSM IV does not diagnose meth use disorder, was it dependence or abuse?
  8. Paragraph starting Line 151: the author should comment on the range of subjective scales used, and when these recordings were done (before/during/after?). Also list the number of studies that had control cues and control groups, and describe the ones that showed more cue meth cue vs control cue reactivity in meth users compared to controls.
  9. Line 212: was it compared to healthy controls? Or neutral cues? Please clarify
  10. Line 215: Was the correlation observed when presented with meth cues?
  11. Line 218: not too sure what is meant by personal attributes?
  12. Line 268 and 270: doses of medication need to be specified.

Discussion

  1. Focus the whole discussion around studies that actually had control cues and control groups.
  2. More critical thinking on the studies should be presented. Were there urine tests to actually corroborate the reduction in cue reactivity was related to people’s drug use behavior? What did the studies with the highest sample size show? Whenever referring to ‘methodological heterogeneity’, give an example of how the different methods influenced the outcome.
  3. Discuss whether the cue reactivity findings are due to cognitive effects of meth (Guerin et al., 2019) and not specific to meth-induced cues?
  4. Discuss whether there are any differences between treatment seeking and non-treatment seeking when the data is available. Abstinence should also be discussed in more depth (active users vs abstinent).
  5. Paragraph starting Line 363: It would be good to state what worked and what didn’t work OR what is associated with cue reactivity and what isn’t
  6. Lines 274-376: As noted previously? Where? Delete.

Minor

  1. Throughout the manuscript: ‘physiologic’ should be spelt ‘physiological’

Guerin, A. A., Bonomo, Y., Lawrence, A. J., Baune, B. T., Nestler, E. J., Rossell, S. L., & Kim, J. H. (2019). Cognition and Related Neural Findings on Methamphetamine Use Disorder: Insights and Treatment Implications From Schizophrenia Research. Frontiers in Psychiatry, 10, 880. http://doi.org/10.3389/fpsyt.2019.00880

Volkow, N. D., Wang, G.-J., Telang, F., Fowler, J. S., Logan, J., Childress, A.-R., et al. (2006). Cocaine cues and dopamine in dorsal striatum: mechanism of craving in cocaine addiction. The Journal of Neuroscience : the Official Journal of the Society for Neuroscience, 26(24), 6583–6588. http://doi.org/10.1523/JNEUROSCI.1544-06.2006

Reviewer 2 Report

The review by Seow, Ong, Hombali, AshaRani, and Subramaniam explores studies that have examined responses to drug-related cues in people with methamphetamine use disorder. This is a comprehensive review that appears to have been conducted using accepted, best-practise scoping review methods. The authors have explained the methods well.

By synthesising findings from a wide range of different types of cue reactivity studies, they make a useful contribution that will be appreciated by psychological scientists and addiction treatment researchers. The sections about pharmacological and brain stimulation modulation of cue reactivity are particularly interesting. The relevance of the findings to future research and treatment is summarised well, with appropriately conservative interpretation. They are well-justified in concluding that cue reactivity research has implications for development of treatments for methamphetamine use disorder. I have little doubt that this review will be highly-cited, and I recommend its publication after some amendment.

It appears the authors did not search clinical trial registries. It is now very easy to search all the World Health Organization-approved clinical trial registries at (https://apps.who.int/trialsearch/) and this should have been part of the search strategy, in case there are additional relevant studies that have been completed but haven’t published findings yet (but which may have uploaded relevant results to the clinical trial registry). The authors should check this and add any additional studies if there are any (there may not be any, but they should check).

There are also numerous minor problems with the quality of writing (e.g., unclear or vague writing) that should be addressed before this is published. I list these in detail below:

The authors make several unreferenced claims in the introduction such as “Drug craving … contributes significantly to the relapse” (line 27) and “cues … can elicit craving and increase the likelihood of repeated drug use” (lines 34-35). Such sweeping, categorical claims need to be referenced. Moreover, in regard to the first sentence, it is my understanding that the contribution of craving to relapse is not necessarily a clear, settled question, and that the relationship between craving and relapse is perhaps dependent on the type of craving measured (e.g., craving in general vs. cue-elicited craving), how it is measured, the population or drug concerned, and how “relapse” is measured. I would encoursge the authors to either engage in more nuanced exploration of this question (if they want to discuss craving in general), or alternatively, focus more specifically on cue-induced craving avoiding more sweeping statements about craving in general.

In lines 49-52, they discuss negative consequences of methamphetamine “use” in a manner that implies these consequences occur with any “use”, when in fact many people who use methamphetamine (particularly infrequent users) experience none of these consequences. Some of these consequences usually only arise from heavy or prolonged use, or are associated with use disorder or certain routes of administration. It would be good if the authors clarified the relationship between levels of use and/or diagnostic categories and these outcomes (e.g., if these outcomes are dose-dependent, related to chronicity, associated with certain use disorder diagnostic categories) rather than just attributing these outcomes to “use” in general.

In lines 52-56 the authors claim that methamphetamine is a global health issue, but only present data for the United States. If they want to claim it is a global issue, rather than just an American issue, they should provide global statistics.

Line 57: They state that methamphetamine is “the 2nd most common illicit drug abused worldwide” with no reference. This should be referenced.

Line 59-60: “Methamphetamine craving … is a cause for concern for its widespread use”. It’s unclear what “cause for concern for its widespread use” is supposed to mean.

Line 61-62: “Research efforts in the area of cue reactivity have been reported to be promising …”. Promising in what way? Explain this better.

The authors use the term “personal attribute” in several places (e.g., Figure 2, lines 218, 219, 366), which is a very general and unclear term. It appears they really mean “personality attribute” or “psychological attribute”, so I would suggest using one of these phrases for better clarity.

The authors state in lines 97-99 that “Studies that … recruited participants with other polysubstance (cocaine, cannabis, heroin etc.) use such that findings specific to methamphetamine cues or users could not be determined, were excluded”. It’s not clear to me whether that means they excluded any study that included any poly-substance users (even if the actual measure was specific to methamphetamine-related cues), or whether they mean they only excluded such studies if there were no measures specific to methamphetamine cues. Could they please explain and re-word to clarify this? If they have excluded any study where participants also used other substances, even if the measure was specific to methamphetamine cue reactivity, I think this is problematic and potentially unjustified, but it’s not clear if that’s what the authors mean.

In lines 126-128, the authors list the various cue modalities used by the studies they review. While I realise that, in the tables, they detail further which studies used which modalities, it would be helpful to summarise in this sentence how many studies used each modality for example by putting “(n= )” after each modality listed, with the number of studies using that modality noted.

There are a number of problems with the terminology used in Table 1. When describing the sample, the authors should use recognised diagnostic terminology (i.e., “meeting criteria for METH dependence/abuse/use disorder) if that is how participants were selected, or otherwise clarify the criterion on which participants were selected. If participants were patients of some type of treatment facility, make this clear in the study sample column, and be consistent about designating whether samples were treatment or non-treatment, currently abstinent or currently using (and, if abstinent, for what minimum and/or maximum durations, if that was specified), etc. Avoid vague, unscientific, and potentially stigmatising terminology like “METH abusers” or “addicts”. Also describe use of drugs as “use”, not “abuse” (e.g., in description of Ekhtiari et al. (2010)) – “abuse” is a diagnostic term for a behavioural pattern associated with sustained use over multiple occasions, not a term for a single episode of use.

The authors are also inconsistent in their use of terms for participants in Table 1, sometimes saying “participant”, other times saying “subjects”, “individuals”, or “patients”. Use “participants” as the default term, unless specifying that they were patients of a treatment facility.

The authors also use other unclear terminology in table 1:

When describing Ekhtiari et al. (2010), does “instruments” mean musical instruments or drug paraphernalia? The acronym “CICT” used in describing the same study is also not explained. Nor is it clear why parentheses are used every time “CICT” is written!

What is “simulated meth”?

In describing Yin et al. (2012), “brain fMRI” can be shortened to “fMRI” – we know which organ fMRI is used to measure. “Pictures of happy, sad” is missing necessary description – pictures of happy and sad what? Faces? Situations? The outcome should be described as “Activation”, not “Abnormal activation”. They are measuring activation whether or not it’s abnormal (which is difficult to define in any case).

Wang et al. (2013): “patient” should be “patients”. Re-word “in random” for clarity – does this mean that which type of cues were presented first was randomised? Or were there multiple session in random order with different types of drug cues?

Lopez et al. (2015): “METH users as assessed by the Structured Clinical Interview for DSM-IV” is nonsense because “METH user” is not a DSM diagnostic category. Also what does “METH use via no specific route of administration” mean? Re-word for clarity.

Malcolm et al. (2016): What does “functional brain imaging” mean? fMRI? Be specific. Also define “rest conditions”. “Days since last use of METH” shouldn’t be listed as an outcome – it’s a correlate or predictor and could be mentioned in the methods column instead. Describe direction of findings – ventral striatum activation correlated with days since last meth use … in what direction? Negatively? Positively?

The description of Shahmohammadi et al. (2016)’s findings is very poorly-worded. Do you mean they had larger P300 amplitudes? Or more positive average voltage in the 300-600 ms epoch? Don’t use vague, unscientific phrases like “brain activities”. Describe results specifically in terms of the actual measure used.

Huang et al. (2018): “fMRI paradigm” should be mentioned in the method column, not the “drug cues” column. Also, what does “simulation scenarios” mean – describe better.

Wang, Shen, & Wu (2018): “physiological EEG” should just be “EEG”. Of course EEG is physiological, so that term is redundant.

Liang et al. (2019): “intake utensils, tools, and … scenarios” – how do “tools” differ from “intake utensils”? Try to be concise and use consistent, clear terminology. Also, clarify whether the attention bias task was a measurement or modification task. Also, the main findings do not measure attention bias anywhere, so it is unclear how attention bias was involved in this study, or whether it needs to be mentioned at all.

Chen et al. (2020): “SCID and DSM” is redundant. SCID is just a diagnostic interview to assess DSM criteria. Just say “DSM-5” (assuming it’s version 5). Also, what were the cut-offs used to define “short” and “long term” abstinence?

To summarise, nearly every row of this table has some unclear, inconcise, inconsistent and/or inappropriate wording. This table needs to be thoroughly proof-read and improved.

Line 153: “increase mood” is far too non-specific to have any useful meaning. What kind of mood is increased?

Line 154: “individuals who used or abstained from the drug” is too vague. By “individuals” do you specifically mean people with a methamphetamine use disorder?

Line 171: “if any” – if any what?

Line 175-176 – describe the direction of the difference (e.g., lower or higher amplitude?), not just the fact that there was a difference.

Lines 198-200: “Ekhtiari et al. (2009) found age of onset of drug abuse to be negatively correlated with level of craving responsiveness. It was proposed that as addiction progresses from an impulsive reward-directed behavior to a compulsive habit, craving responsiveness decreases.” The second sentence isn’t logically linked with the first. It would make sense if the predictor was current age, or duration since onset, but the predictor is age of onset. And if higher age of onset is associated with reduced cue-induced craving, doesn't this imply (assuming that earlier age of onset is related to longer duration since onset) that longer duration of meth use is associated with higher craving?

Lines 226-227: define “short-term” and “long-term”.

Line 239: should “proposed” be “found”? I.e., did they just “propose” that this relationship could exist or did they actually measure and report this as a result? I assume it’s actually the latter.

Table 2 is much better-written than Table 1, but there are still a few things that should be addressed:

They describe Roche et al.’s (2016) sample as “individuals who met DSM-IV criteria for METH use disorder”. The DSM-IV still divided use disorder between dependence and abuse. Please clarify which disorder participants needed to meet criteria for (or if they could meet criteria for either, state this explicitly).

Regarding De Santis et al. (2009), it’s not clear to me why it’s relevant to describe the physiological and craving outcomes in the “outcomes” column, if those outcomes weren’t actually reported. Also, meth use in the past 90 days is a correlate/predictor, not an outcome, so mention it in the methods column. Moreover, it’s really not clear to me what was done in this study. Was the cue exposure used purely as an intervention? If results on reactivity to these cues were not reported, then it’s really not clear to me why it would be relevant to include this in the review.

Price et al. (2010): “Heart rate and skin conductance patterns did not mirror that of craving ratings”. So what did happen with these measures? Describe what they found, not just what they didn’t find.

Where participant groups are described as “abstinent” (e.g., Shahbabaie et al., 2014), please detail (if known) what the minimum and/or maximum time abstinent was – there are potentially important differences between people abstinent for years vs. people abstinent for a few days.

Lopez et al. (2015): “METH smokers as assessed by the SCID DSM-IV”. “Meth smoker” is not a DSM-IV diagnostic category. What was the actual diagnostic criterion used?

Su et al. (2017): “Changes in craving ratings were also significantly predicted by age and education.” Describe the direction of the effects.

Anaraki et al. (2019): what does “instant craving” mean?

Wang, Liu, & Shen (2019): Please explain what the “Meth-using” VAS measures.

Lines 268, 298-299, and 304-305 – when noting negative findings (e.g., lack of effect of aripiprazole, tDCS, or approach bias modification) from very small studies, I think the authors should mention that the sample sizes were very small, since the elevated possibility of type 2 errors qualifies interpretations of these findings.

Lines 277-286: The authors contrast conflicting findings from different studies regarding 1 Hz TMS. Were there any notable methodological differences (e.g., in terms of how craving was elicited, nature of population, etc.) between these studies? If so, these should be noted, given their possible influence on the findings.

Line 284: “bilateral hemisphere” could be phrased better, e.g., “left vs. right hemisphere”.

Line 297: what does “instant subjective cravings” mean? Re-word for clarity.

Lines 308-309 say “28 days”, but the table says 14 days – please clarify which one is correct.

Lines 364-365: The authors list demographic factors among those that “affect reactivity to methamphetamine-related cues”, but earlier (e.g., lines 190-194) argue that these factors are generally NOT associated with cue reactivity, so this is an inconsistent interpretation that needs to be changed.

Lines 367-369: The authors list several interventions, implying they have all been shown to affect methamphetamine cue reactivity, but this contradicts earlier sections where they note that there is evidence that some of these interventions, but not others, alter cue reactivity. This paragraph needs to be re-written to better distinguish between factors for which there is evidence of an effect, factors where there is evidence of no effect, and factors where (due to inadequate or only small studies), the question is still open.

Line 370-372: The closing sentence of this paragraph is very vague. Be more specific about what you mean.

Additional minor points (e.g., possible typos):

There are a few typos in Figure 2 (e.g., spaces where there shouldn’t be, or absence of spaces where there should be).

Line 183: should “distracting” be “distraction”?

Line 307: should “extinct” be “extinguish”?

Line 336: should “incubation” be “induction”?

Line 437: should end with a “.” Not a “

I noticed a few bits of strange formatting in the reference list that I thought might be typos (e.g. references 2, 3, 4, 10)

Line 437: “Journal of Addiction Research & Therapy S” – is the “S” a typo?

Line 554 is the “%” a typo?

Round 2

Reviewer 1 Report

I appreciate the corrections, especially in the tables. Many concerns were reasoned against rather than addressed, most of which I can accept. Unfortunately, there were a few important concerns that have not been addressed (nor reasoned against sufficiently) that still need to be addressed. None of these are new concerns but highlighted in the first review.  

  1. Rationale for 1995 cut off should be added into the manuscript, not just stated in the rebuttal.
  2. It is strange why the tables need to be presented in a similar format. They are vastly different types of studies and there are plenty of influential reviews with different table formats to cater towards different types of information Clinical trials vs basic science studies provide. In any case, the revised tables are much more informative but what ‘neutral’ cues are should be stated somewhere (don’t have to be in the table, but somewhere in text e.g., ‘neutral cues were typically …’). Were they nature cues? Neutral face expressions? Objects (like squares and triangles)? Such info is well within the scope of a scoping review. 
  3. Discussion – lack of control groups etc should be explicitly stated in the general discussion as a shortcoming for many studies because it may be the case that non-meth users have similar cue reactivity to meth cues (which are much more stimulating and confronting than neutral cues) – this is pretty much the most informative thing that this scoping review can conclude. That many of these studies whether cue reactivity is indeed a result of pavlovian conditioning that leads to expectation of drug availability is still unclear.

Minor

4. Line 75: "Despite amphetamine-type stimulants (mainly methamphetamine) being the second most widely used class of drugs worldwide [21]" – I think the authors mean “class of ‘illicit’ drugs”.

5. Don’t forget to revise “meth addicts" for Yin et al 2012 in table 1.
